# Preclinical Efficacy of a Capsid Virus-like Particle-Based Vaccine Targeting IL-1β for Treatment of Allergic Contact Dermatitis

**DOI:** 10.3390/vaccines10050828

**Published:** 2022-05-23

**Authors:** Louise Goksøyr, Anders B. Funch, Anna K. Okholm, Thor G. Theander, Willem Adriaan de Jongh, Charlotte M. Bonefeld, Adam F. Sander

**Affiliations:** 1Centre for Medical Parasitology, Department for Immunology and Microbiology, Faculty of Health and Medical Sciences, University of Copenhagen, 2200 Copenhagen, Denmark; louiseg@sund.ku.dk (L.G.); anna.okholm@sund.ku.dk (A.K.O.); thor@sund.ku.dk (T.G.T.); 2AdaptVac Aps, 2200 Copenhagen, Denmark; wdj@adaptvac.com; 3LEO Foundation Skin Immunology Research Center, Department for Immunology and Microbiology, Faculty of Health and Medical Sciences, University of Copenhagen, 2200 Copenhagen, Denmark; abfunch@sund.ku.dk (A.B.F.); cmenne@sund.ku.dk (C.M.B.)

**Keywords:** vaccine, virus-like particle, AP205, IL-1β, allergic contact dermatitis

## Abstract

Hypersensitivity to a contact allergen is one of the most abundant forms of inflammatory skin disease. Today, more than 20% of the general population are sensitized to one or more contact allergens, making this disease an important healthcare issue, as re-exposure to the allergen can initiate the clinical disease termed allergic contact dermatitis (ACD). The current standard treatment using corticosteroids is effective, but it has side effects when used for longer periods. Therefore, there is a need for new alternative therapies for severe ACD. In this study, we used the versatile Tag/Catcher AP205 capsid virus-like particle (cVLP) vaccine platform to develop an IL-1β-targeted vaccine and to assess the immunogenicity and in vivo efficacy of the vaccine in a translational mouse model of ACD. We show that vaccination with cVLPs displaying full-length murine IL-1β elicits high titers of neutralizing antibodies, leading to a significant reduction in local IL-1β levels as well as clinical symptoms induced by treatment with 1-Fluoro-2,4-dinitrobenzene (DNFB). Moreover, we show that a single amino acid mutation in muIL-1β reduces the biological activity while maintaining the ability to induce neutralizing antibodies. Collectively, the data suggest that a cVLP-based vaccine displaying full-length IL-1β represents a promising vaccine candidate for use as an alternative treatment modality against severe ACD.

## 1. Introduction

Hypersensitivity to a contact allergen is one of the most abundant forms of inflammatory skin disease. Notably, at least 20% of the general population is sensitized to one or more contact allergens, with a higher prevalence in women [1,2,3,4]. The response to contact allergens can be divided into sensitization and elicitation phase [5,6]. The initial sensitization phase occurs when the contact allergen penetrates the skin and initiates the expansion of allergen-specific T-cells. Subsequent re-exposure to the allergen initiates the elicitation phase, which can lead to the development of allergic contact dermatitis (ACD) characterized by redness, edema, eczema, and blister formation on the skin [7,8]. Currently, ACD is primarily treated with corticosteroids. However, despite being very efficient, this therapy has several side effects when used for longer periods, including the risk of causing skin allergy [9]. Consequently, there is a need for alternative and more specific treatments. Interleukin-1β (IL-1β) is recognized as a central player in the immune response following allergen exposure [10,11,12]. Specifically, a study in mice showed that following allergen exposure, the earliest change in the cytokine profile was an increase in IL-1β mRNA levels, which could be measured within the first 15 min [13,14]. Further in vivo studies have underlined the central role of IL-1β in relation to ACD by showing that injection of anti-IL-1β blocks the challenge response to the allergen trinitrochlorobenzene (TNCB), whereas administration of anti-IL-1α only had a minor effect [15]. Thus far, no ACD treatments exist that directly target IL-1β. However, a recombinant version of the IL-1 receptor antagonist (IL-1Ra) (Anakinra, Kineret^®^, Swedish Orphan Biovitrum AB, Stockholm, Sweden) has shown to be effective in reducing symptoms of other inflammatory diseases, such as rheumatoid arthritis (RA). However, this treatment requires daily administration due to the short half-life of the drug [16]. Alternatively, the IL-1 pathway can be blocked by monoclonal antibody (mAb) therapy. Nevertheless, mAb therapy has significant limitations, both in terms of patient access and clinical utility. Frequent administration of high doses is associated with high costs, as well as the risk of developing anti-drug immunity [17,18]. Thus far, only one anti-IL-1β mAb (Canakinumab, Ilaris^®^ (Novartis, Basel, Switzerland)) has been approved for the treatment of a couple of rare autoinflammatory diseases [19,20,21]. Active vaccination represents another promising alternative treatment modality. However, for induction of therapeutic antibody (Ab) responses against disease-associated self-antigens, vaccines must effectively overcome B-cell immune tolerance [18,22]. Traditionally, this has been accomplished by genetic fusion with a carrier protein containing foreign T-helper cell epitopes [18]. However, limited induction and rapid Ab decline suggest suboptimal B-cell activation [18,23]. Since then, it has been shown that multivalent, high-density, and preferred unidirectional antigen display is important for obtaining a strong vaccine-induced Ab response [24,25,26]. Here, virus-like particles (VLPs) have been shown to be promising vaccine platforms. The vaccination of mice with IL-1β displayed on keyhole limpet hemocyanin (KLH) or VLPs has been shown to induce therapeutic antibodies for the treatment of RA [27,28] and type 2 diabetes [29], respectively. Moreover, this VLP-based IL-1β vaccine was also shown to be safe and well tolerated in humans [30].

We have developed a highly versatile Tag/Catcher AP205 capsid VLP (cVLP) vaccine technology [31,32]. Through genetic engineering, the capsid protein from the *Acinetobacter phage* AP205 has been fused to a split protein binding partner (i.e., Tag or Catcher), enabling the surface exposure of 180 binding partners on the assembled cVLP. This allows for the subsequent conjugation of antigens genetically fused to the corresponding binding partner, providing unidirectional and high-density antigen display for strong B-cell activation [31]. In the present study, we show that active immunization of mice with cVLPs displaying full-length (FL) murine IL-1β results in a strong induction of antigen-specific neutralizing antibodies, which leads to a significant reduction of the clinical signs measured following the challenge reaction to the allergen 1-Fluoro-2,4-dinitrobenzene (DNFB). Finally, we show that a single amino acid mutation in muIL-1β reduces the biological activity while maintaining the ability to induce high quality and effective antibodies. Consequently, these collective data support the further clinical development of an IL-1β targeting vaccine for the treatment of ACD.

## 2. Materials and Methods

### 2.1. Design, Expression, and Purification of Catcher-cVLP

The *Acinetobacter phage* AP205, displaying an N-terminal Catcher per subunit, was designed and produced as previously described [31]. Purified cVLPs were dialyzed O/N at 4 °C into PBS, pH 7.4 using cutoff 1000 kDa (SpectrumLabs, San Francisco, CA, USA). Total protein concentration was determined by Bicinchoninic acid (BCA) assay (Thermo Fisher Scientific, Waltham, MA, USA).

### 2.2. Design, Expression, and Purification of Recombinant IL-1β Proteins

Murine IL-1β comprising amino acids 118–269 (Gene ID: 16176) was designed with an N-terminal 6xHis-tag followed by a TEV cleavage site, and a C-terminal binding Tag. Additionally, a flexible GGS linker was inserted between IL-1β and the binding tag gene sequence. Flanking NcoI and NotI restriction sites were added to the N- and C-terminus, respectively. The final gene sequence was codon optimized for expression in *E. coli* and synthesized by GeneArt (Thermo Fisher Scientific). Mutations (i.e., R11G, Q15G, H30G, N32G) were introduced into the muIL-1β-Tag sequence by overlap-extension PCR (98 °C for 10 s, 25–35 × (98 °C for 10 s, 55–58 °C for 20 s, and 72 °C for 30 s/kb), 72 °C for 5 min). The primers used for the PCR reactions are listed in Appendix A. The general 3′ forward and 5′ reverse primers annealed to the N- and C-terminus of the His-TEV-muIL-1β-Tag gene sequence, respectively, are not provided. Gene sequences were cloned into a pET15-b vector and transformed into One Shot^®^ BL21 Star™ (DE3) cells (Thermo Fisher Scientific, Waltham, MA, USA). Expression was performed in 2xYT media at 37 °C. Protein expression was induced at OD_600_ = 0.6–0.8 by addition of IPTG to a final concentration of 0.1–1 mM for IL-1β–Tag constructs, followed by incubation at 20 °C for 16–18 h. Recombinant muIL-1β constructs were purified by IMAC (PBS, pH 7.4 with either 60 mM or 500 mM imidazole for binding and elution, respectively) and size exclusion chromatography (SEC) using a HiLoad Superdex 75 pg column (GE Healthcare, Chicago, IL, USA) (PBS, pH 7.4) and an ÄKTAexpress purification system (GE-Healthcare).

### 2.3. Vaccine Formulation

cVLPs and muIL-1β antigens were purified for lipopolysaccharides (LPS) prior to vaccine formulation, as previously described using Triton X-114 [33]. To formulate the cVLP:IL1β vaccines, the Catcher-cVLP and Tag-IL1β antigens were mixed in a 1:2 molar ratio and incubated O/N at 4 °C. Excess unbound antigen was removed by density gradient ultracentrifugation using an Optiprep™ (Sigma-Aldrich, St Louis, MO, USA) Step gradient (23, 29, and 35%) [31]. Purified cVLP:IL-1β vaccines were dialysed against sterile PBS, pH 7.4, using a cutoff of 1000 kDa (SpectrumLabs). All vaccines were subjected to a centrifugation stability test (16,000× *g* for 2 min). An equal number of pre- and post-spin samples were loaded on SDS-PAGE to assess potential precipitation, indicating vaccine aggregation. Antigen concentration on the cVLP and coupling efficiency (calculated as cVLP subunits/coupled cVLP subunits + uncoupled cVLP subunits) were estimated by densitometry from SDS-PAGE gels using ImageQuant TL (Cytiva, Marlborough, MA, USA).

### 2.4. Quality Assessment of cVLP:IL-1β Vaccines

Purified cVLP:IL-1β vaccines were quality checked by Dynamic Light Scattering (DLS) and negative stain transmission electron microscopy (TEM). For DLS analysis, vaccines were spun at 16,000× *g* for 2 min (Eppendorf tabletop centrifuge 5424 R) before being loaded into a disposable cuvette. The samples were run with 20 acquisitions for 7 s at 25 °C using a DynaPro Nanostar (Wyatt Technologies, Santa Barbara, CA, USA). The estimated diameter of the cVLP:IL-1β particle population and the percent polydispersity (%Pd) were calculated using Wyatt DYNAMICS software (7.10.0.21) (Wyatt Technologies). For TEM, vaccines were adsorbed onto 200-mesh carbon-coated grids and stained with 2% uranyl acetate for 1 min. The excess stain was removed with filter paper. The grids were analysed using a CM 100 BioTWIN electron microscope (Philips, Amsterdam, Netherlands).

### 2.5. Biological Activity of IL-1β Antigens Using HEK-Blue Cells

HEK-Blue™ IL-1R Cells (Invitrogen, Waltham, MA, USA) were passaged in DMEM GlutaMAX™ (Gibco, Waltham, MA, USA), 2 mM L-glutamine, 10% heat-inactivated FBS (Gibco), 100 U/mL penicillin, 100 µg/mL streptomycin, HEK-blue™ selection (Invivogen, San Diego, CA, USA). These cells express both human and murine IL-1R, thus binding of human and murine IL-1β activates downstream NF-κB and AP-1 pathways, resulting in expression of a SEAP reporter gene. For the biological activity assay, IL-1β constructs were prepared in a 3-fold dilution in test media (i.e., growth media without HEK-blue™ selection) starting from 100 ng/mL. For control samples,10 ng/mL huTNFα (negative control, Abcam, Cambridge, UK), huIL-1β (positive control, Abcam), or muIL-1β (positive control, Nordic BioSite, Täby, Sweden) were added to the plate in duplicates. Cells were prepared in 3 × 10^5^ cells/mL suspension in test media, and 180 µL cell suspension was added to a flat-bottom 96-well plate (Nuclon™ Delta surface, Thermo Fisher Scientific) (~50.000 cells/well). Sample and control samples (20 µL) were added to the plate, followed by O/N incubation (~16 h) at 37 °C, 5% CO_2_. Supernatant (20 µL) was transferred to a new flat-bottom 96-well plate and mixed with 180 µL QUANTI-blue™ solution (Invivogen). Plates were incubated for 3 h at 37 °C, 5% CO_2_. SEAP levels were measured at 620 nm using a HiPo MPP-96 microplate photometer (BioSan, Riga, Latvia).

### 2.6. Mouse Immunization Studies

Experiments were authorized by the National Animal Experiment Inspectorate (Dyreforsøgstilsynet, license no: 2018-15-0201-01541) and performed according to national guidelines. 6–8 weeks old female C57BL/6 mice (Janvier, Denmark) were immunized intramuscularly (I.M.) in a prime-boost-boost regime with either 2, 7, or 10 µg IL-1β displayed on cVLPs. All vaccines were formulated in Addavax™ (Invivogen). Blood samples were collected 2 weeks after each immunization, as well as 17 and 22 weeks after the prime immunization for the dose-dependence study. Blood samples were taken according to the size of the mice (~50–100 µL). To obtain the serum sample, blood samples were stored O/N at 4° following two cycles of centrifugation at 800× *g* for 8 min at 8 °C.

To induce contact hypersensitivity (CHS), mice were sensitized for three consecutive days (day 0–2) on the dorsal side of both ears with 25 µL 0.15% 1-Fluoro-2,4-dinitrobenzene (DNFB) in a 1:4 olive oil:acetone (OOA) mixture [34]. Mice were immunized in a 3-week interval prime-boost-boost regime (days 14, 35, and 56). On day 77 (3 weeks after 2nd boost immunization), mice were challenged on the dorsal side of both ears with 25 µL 0.15% DNFB formulated in OOA. Control mice were exposed to OOA or DNFB/OOA on day 0–2 and day 77. Mice were euthanized 24 or 96 h after the challenge. Ear thickness was measured on both ears using an electronic digital micrometer (Mitutoyo Corporation, Sakado, Japan, Model: PK-1012CPX). To measure the IL-1β level, mice were terminated 24 h after the challenge, and the ears were collected. The dorsal and ventral sides of the ears were separated before the dorsal side was frozen in liquid nitrogen and used for further analysis.

### 2.7. Analysis of Vaccine-Induced Antibody Responses

Antigen-specific total IgG titers were measured by ELISA. 96-well plates (Nunc MaxiSorp, Invitrogen) were coated overnight at 4 °C with 0.1 µg/well Tag-muIL-1β in PBS, pH 7.4. Plates were blocked with 0.5% skimmed milk in PBS O/N at 4 °C. Mouse serum was diluted in a 3-fold dilution starting from 1:200, followed by incubation for 1 h at RT. Plates were washed 3 times in PBS in between steps. Total serum IgG was detected using Horseradish peroxidase (HRP) conjugated goat-anti mouse IgG (Invitrogen)diluted 1:1000 in blocking buffer and incubated for 1 h at RT. Plates were developed with TMB X-tra substrate (Kem-En-Tec, Taastrup, Denmark) and the absorbance was measured at 450 nm. For the avidity assay, plates were run in duplicates, followed by a 5 min PBS or 8M urea wash, before incubation with the secondary antibody.

### 2.8. IL-1 Subfamily Cross-Reactivity Western Blot Analysis

For Western blot analysis, 0.1 µg huIL-1β (Abcam), muIL-1β (Nordic BioSite), muIL-1α (Nordic BioSite), muIL-33 (BioLegend, San Diego, CA, USA), muIL-1Ra (R&D Systems, Minneapolis, MN, USA), and Catcher-AP205 cVLP were loaded on SDS-PAGE and transferred onto a nitrocellulose membrane. Membranes were blocked in 5% skimmed milk in TBS-T O/N at 4 °C. Blots were incubated with pooled serum from vaccinated mice diluted 1:300 for 1 h at RT, followed by goat anti-mouse IgG HRP-conjugated (Invitrogen) diluted 1:1000 for 1 h at RT. In between steps, blots were washed three times with TBS-T. Blots were developed using a TMB blotting substrate (Kem-en-Tech).

### 2.9. Quantification of Interleukins in Vaccinated Mice

IL-1β, IL-6 and TNFα were quantified from full ear skin samples (dermis + epidermis) following allergen re-exposure. The skin was removed from the cartilage tissue using surgical tweezers, and snap-frozen in liquid nitrogen 24 h post challenge. Samples were submerged in 800 µL lysis buffer (50 mM Trizma base, 250 mM NaCl, 6.4 mM EDTA, 17.6 mM Triton X-100, pH 7.4, including protease inhibitor cocktail (cOmplete, Roche Diagnostics GmbH, Berlin, Germany)). Tissue homogenization was performed in MK28 hard grinding precellys tubes (Bertin Technologies, Montigny-le-bretonneux, France) using a Precellys Evolution instrument (Bertin Technologies). Protein concentration in the supernatant was quantified by a Bradford assay and stored at −80 °C. The protein concentration of all samples was adjusted to 2 mg/mL. IL-1β levels were analyzed using a mouse IL-1β/IL-1F2 DuoSet ELISA kit (R&D systems), on 10× diluted samples. IL-6 and TNFα levels were analyzed using respective ELISA MAX™ standard kits (Biolegend) on undiluted samples. All assays were performed following manufacturer’s protocol.

### 2.10. Statistical Analysis

Statistical analysis was performed on log-transformed values using one-way ANOVA, Tukey’s multiple comparisons test (adjusted *p*-value < 0.05 was accepted as significant). Differences between cytokine levels were conducted according to the following strategy: (1) to compare the OOA and DNFB control groups, (2) to compare the DNFB control group to the control vaccine group (i.e., unconjugated cVLP), and (3) to compare the DNFB control group to groups vaccinated with cVLP:IL-1β vaccines. All statistical analyses were performed using GraphPad Prism (9.3.1) (GraphPad, San Diego, CA, USA).

## 3. Results

### 3.1. IL-1β Antigen Design

To investigate the biological effect of a murine IL-1β vaccine in a mouse model of ACD, FL murine IL-1β (aa118-269) (Gene id: 16176) was designed with an N-terminal 6xHis-tag followed by a TEV site. Additionally, the IL-1β construct was C-terminally fused to a split-protein Tag, separated by a flexible linker. Several specific mutations have previously been reported to reduce the biological activity of IL-1β while maintaining the affinity for the interleukin-1 receptor 1 (IL-1R1) [35,36,37,38,39]. Theoretically, such mutations could serve to improve the safety of an IL-1β vaccine by decreasing the risk that the antigen could activate signaling by binding to the receptor. For that reason, four selected residues (i.e., R11G, Q15G, H30G and N32G) in the muIL-1β sequence were mutated to Glycine [36,37,38]. These mutant constructs shared a similar overall design to the previously described FL IL-1β antigen. An overview of the designed antigen constructs is presented in Figure 1a. After recombinant expression and purification, the biological activity of both native and mutated muIL-1β antigens was evaluated. Specifically, to enable investigation of IL-1β activation through the NK-κB and AP-1 pathways (and subsequent detection of secreted embryonic alkaline phosphatase (SEAP)), a TLR3, TLR5, and TFNR1-knockout HEK293 cell line, expressing both the human and murine IL-1R1, was used. This cell activation assay indicated that the biological activity of the WT IL-1β-Tag antigen was slightly lower than that of commercial recombinant murine and human IL-1β used as positive controls (Figure 1b and Appendix A). Antigens carrying the R11G, Q15G, or H30G mutation all showed lower biological activity than the WT IL-1β-Tag construct, whereas the N32G mutation did not lower the biological activity (Figure 1b). Since the Q15G mutation had the most profound effect on biological activity, we continued with the IL-1β-Tag WT and Q15G antigens for further comparative formulation and immunogenicity studies.

### 3.2. Vaccine Characterization and Quality Assessment

Vaccine formulations were made by mixing IL-1β antigens with AP205 cVLPs in a 2:1 molar ratio (Figure 2a). Upon mixing, the split-protein Catcher and Tag binding partners (i.e., genetically fused to the cVLP and antigen, respectively) react to form a covalent bond, which results in unidirectional display of the antigen at high density on the cVLP surface (Figure 2a). Covalent coupling of IL-1β antigens to the cVLP was confirmed by SDS-PAGE analysis by the appearance of a protein band of 48 kDa, corresponding to the added size of the IL-1β-Tag antigen (21 kDa) and Catcher-cVLP (27 kDa) (Figure 2b). To assess vaccine stability, an initial centrifugation test (16,000× *g*, 2 min) was conducted to measure potential precipitation. For both vaccine formulations, this did not lead to any detectable loss in the concentration of cVLP:antigen complexes in suspension (i.e., 48 kDa protein coupling band), indicating stable vaccine formulations (Figure 2b). The antigen coupling efficiency (i.e., the percentage of total binding sites per cVLP conjugated to an IL-1β antigen) was estimated to be ~67% for both the cVLP:IL-1β and cVLP:IL-1β Q15G vaccines. The presence of excess antigens in the coupling reactions (i.e., band at 21 kDa) indicates complete surface decoration of the cVLP (Figure 2b).

The cVLP:IL-1β vaccines were further quality assessed by dynamic light scattering (DLS) and transmission electron microscopy (TEM). For both the cVLP:IL-1β (Figure 3a) and cVLP:IL-1β Q15G (Figure 3d) vaccine, DLS shows one main population of monodisperse particles (%polydispersity (%Pd) < 15) with a diameter of approximately 50 nm. However, the cVLP:IL-1β vaccine shows a small population of higher molecular weight (>1000 nm), indicating a higher propensity for aggregation. The presence of intact cVLP:IL-1β particles of the expected size was confirmed by TEM (Figure 3b,c,e,f).

### 3.3. Immunogenicity of cVLP:IL-1β Vaccines

The immunogenicity of the cVLP:IL-1β vaccines was assessed in female C57BL/6 mice vaccinated intramuscularly (I.M.) in a three-week interval prime-boost-boost regimen (Figure 4a). All vaccines were formulated in Addavax™ (Invivogen). An initial antigen dose-escalation study was performed on the cVLP:IL-1β vaccine, in which mice received 2, 7, or 10 µg IL-1β antigen displayed on cVLPs (Figure 4b). Blood samples were taken two weeks after each immunization, as well as after 17 and 22 weeks, to follow the kinetics of the vaccine-induced Ab response (Figure 4a). Antigen-specific IgG titers were measured by ELISA using recombinant IL-1β-Tag as a capture antigen. The ELISA capture antigen was tested head-to-head against a commercial murine IL-1β protein, which showed no significant difference (Appendix A). Vaccination with cVLP:IL-1β led to seroconversion in all mice, and the antigen-specific IgG titers were similar in the tested dose-interval (Figure 4b and Appendix A). Thus, for further head-to-head testing of the cVLP:IL-1β and cVLP:IL-1β Q15G vaccines, the 2 µg antigen dose was used (Figure 4c). Similar to the first study, all mice vaccinated with cVLP:IL-1β and cVLP:IL-1β Q15G seroconverted (Figure 4c and Appendix A). A significant difference in the level of IL-1β-specific IgG titers was observed between the two vaccines at week 2 (W2) (*p* < 0.0001) and week 5 (W5) (*p* = 0.0249), with the cVLP:IL-1β Q15G inducing a higher total IgG titer, despite the coat being the WT IL-1β sequence. However, this effect was not significant after the second boost at week 8 (W8) (Figure 4c). Avidity assays can be used to assess the binding strength of vaccine-induced Abs, where low-avidity Abs are removed by the introduction of an 8M urea wash in the ELISA protocol. Specifically, dilutions of serum from vaccinated mice (W8) were either subjected to a PBS or 8M urea wash. The avidity index represents the effect of urea on antibody–antigen binding, calculated as the area under the curve (AUC) after urea exposure, divided by the AUC without urea exposure (i.e., PBS). These data show that the avidity index was similar for mice vaccinated with either of the cVLP:IL-1β vaccines, despite lower total IgG titers for mice vaccinated with the cVLP:IL-1β vaccine (Figure 4d). For both vaccines, approximately 25% of vaccine-induced Abs remained bound after urea exposure, indicating a high binding affinity (Figure 4d).

### 3.4. IL-1 Family Member Cross-Reactivity of Vaccine-Induced Anti-IL-1β Antibodies

To investigate to what extent vaccine-induced antibodies cross-react with other members of the IL-1 subfamily, the reactivity of serum from vaccinated mice was tested against huIL-1β, muIL-1β, muIL-1α, muIL-33 as well as muIL-1Ra in a western blot. Catcher-cVLP was used as a positive control (Figure 5). Pooled serum from mice vaccinated with cVLP:IL-1β and cVLP:IL-1β Q15G reacted with human and murine IL-1β (17 kDa) (Figure 5, lane 1–2), as well as the positive control (Figure 5, lane 6). By contrast, these serum pools did not react with muIL-1α, muIL-33 or muIL-1Ra (Figure 5, lane 3–5), indicating that the vaccine-induced antibodies do not cross-react with other members of the IL-1 subfamily.

### 3.5. Vaccination against IL-1β Inhibits the Challenge Response to DNFB

The biological effect of vaccine-induced anti-IL-1β antibodies was tested in a mouse model of ACD. Specifically, mice were sensitized on the ears with DNFB for three constitutive days before being vaccinated (prime-boost-boost) with either cVLP:IL-1β, cVLP:IL-1β Q15G, or with a control vaccine (i.e., unconjugated Catcher-cVLP). As a negative control for the model, the mice were exposed to olive oil: acetone (OOA, the vehicle for DNFB). Three weeks after the final vaccination (D77), the mice were challenged with DNFB on the ears and divided into two major groups. In the first group, the clinical progression was followed by measurement of the ear thickness during the following 96 h, while the other group of mice was sacrificed to measure local cytokine concentrations in the ear tissue (Figure 6a).

The challenge study showed a peak response in ear thickness for DNFB-treated mice, and mice vaccinated with unconjugated cVLP 24 h post challenge, corresponding to an increase of ~25% (Figure 6b). Mice treated with OOA showed no significant changes in ear thickness (Figure 6b). By contrast, DNFB-treated mice vaccinated with either the cVLP:IL-1β or cVLP:IL-1β Q15G vaccine maintained a similar ear thickness to that of the OOA-treated mice, which was significantly (*p* < 0.0001) thinner than that of DNFB-treated mice (Figure 6b and Appendix A). The in vivo efficacy of these vaccines was further supported by measuring IL-1β levels in inflamed ear tissue. Furthermore, to assess whether neutralization of IL-1β by vaccine-induced Abs influenced other important pro-inflammatory cytokines, the levels of IL-6 and tumor necrosis factor alpha (TNFα) were also measured in the ear tissue. As expected, the DNFB challenge resulted in increased cytokine levels in the ear tissue. Compared to the non-sensitized mice (i.e., OOA treated group), the ears from sensitized unvaccinated mice contained markedly higher levels of IL-1β and IL-6 (*p* < 0.0001), whereas the TNFα levels were about twice as high (*p* = 0.0149) (Figure 6c–e). However, ears from mice vaccinated with the cVLP:IL-1β and the cVLP:IL-1β Q15G vaccines contained statistically significant lower levels of IL-1β, as compared to unvaccinated mice (mean ± SEM from unvaccinated mice 569 ± 108 pg/mL, and 151 ± 32 pg/mL (*p* = 0.0082) and 15 ± 8 pg/mL (*p* < 0.0001) for mice vaccinated with cVLP:IL-1β and the cVLP:IL-1β Q15G, respectively) (Figure 6c). Interestingly, the IL-1β levels in ears from mice vaccinated with cVLP:IL-1β Q15G was significantly lower than compared to mice vaccinated with cVLP:IL-1β (*p* < 0.0001) (Figure 6c). Of importance, the levels of IL-6 and TNFα were similar in the ears of unvaccinated mice and mice vaccinated with either of the cVLP:IL-1β vaccines (Figure 6d,e). Together, these data show that vaccination with either of the cVLP:IL-1β vaccines resulted in a reduction of local IL-1β levels as well as clinical symptoms induced in a mouse model of ACD, without reducing the local levels of other pro-inflammatory cytokines.

## 4. Discussion

Many consumer products, such as soap and cosmetics, contain several ingredients that cause skin allergies [3,9]. Consequently, more than 20% of the general population is sensitized to one or more contact allergens, which constitute an important health problem [1,4]. The current standard treatment with corticosteroids is effective, although in cases of severe and prolonged disease, more efficient, long-lasting, and well-tolerated therapies are required to ensure skin regeneration. Here, active vaccination targeting IL-1β could serve as a promising alternative to the current standard treatment for patients with severe ACD.

For the development of therapeutic vaccines targeting self-antigens, a fundamental challenge has been to overcome inherent B-cell immune tolerance [18]. To this end, the high-density display of self-antigens on VLPs represents a highly efficient strategy for the induction of strong autoantibody responses in both mice and humans [30,40,41,42,43]. Importantly, this approach is highly dependent on the quality of the antigen display and has long been restricted to peptide antigens [24,26,44]. In this regard, we utilized the modular Tag/Catcher AP205 cVLP vaccine platform to display the FL IL-1β for the induction of broadly neutralizing Abs. This versatile technology enables unidirectional, high-density display of even large and complex antigens and has recently been used for the development of a SARS-CoV-2 vaccine, which has proven safe and highly immunogenic in human phase I/II clinical studies (clinical trial ID: NCT04839146 and NCT05077267) [44]. Moreover, this platform has been used to develop a HER2+ breast cancer vaccine, which demonstrated high efficacy in a preclinical mouse model of spontaneous HER2+ breast cancer, due to strong induction of antigen-specific autoantibody responses [40]. On that basis, we utilized this vaccine technology to test the immunogenicity and in vivo efficacy of cVLP:IL-1β vaccines in a model of ACD. Overall, this study supports the above results by demonstrating that cVLP-display of IL-1β results in the induction of high levels of antigen-specific autoantibodies in mice. In fact, peak anti-IL-1β Ab titers were reached after just two immunizations with 2 µg of antigen, and neither booster immunization nor higher antigen doses resulted in a further increase in the antibody response. When it comes to vaccines targeting cytokines, there is a potential risk that the vaccine antigen can activate the cytokine receptor upon vaccination [45]. However, vaccination of mice with the Fel d1 cat allergen displayed on VLPs could induce a strong neutralizing Ab response without activating mast cells [45,46,47], and several studies of cytokine-based VLP vaccines have further shown that induction of anti-cytokine autoantibody responses can be elicited without concomitant induction of an inflammatory response [28,29,30,48,49]. Thus, these results suggest that in cases where an allergen is used as a vaccine antigen, a VLP display can prevent the vaccine antigen from activating the native receptor of the allergen during vaccination [45,46,47]. Still, to mitigate this potential risk, we introduced the Q15G mutation to the murine IL-1β sequence. For human IL-1β, the Gln15 residue has been shown to be a critical residue in the receptor:ligand binding interface [36], which upon mutation (e.g., Q15G) leads to a complete loss of binding to the IL-1R1 receptor, without causing major structural changes in the IL-1β protein [36,37,38]. As expected, the introduction of the Q15G mutation in the muIL-1β antigen reduced the binding activity to IL-1R1 while retaining the capacity for inducing anti-IL-1β neutralizing IgG Abs.

Based on these fundamental results, the cVLP:IL-1β and cVLP:IL-1β Q15G vaccines were tested head-to-head in a mouse model of ACD. The immunological response to DNFB in this mouse model has previously been shown to be translational to the response found in patients [50]. We showed that immunization with both tested cVLP:IL-1β vaccines resulted in a significant reduction in the clinical signs of inflammation (i.e., measured as ear thickness) following the DNFB challenge. This result was further supported by data showing a significant reduction in local IL-1β levels in the ear tissue of vaccinated mice. Interestingly, mice vaccinated with the cVLP:IL-1β Q15G vaccines showed significantly lower levels of local IL-1β in ear tissue compared to mice vaccinated with the cVLP:IL-1β vaccine. It could be speculated that the greater reduction of IL-1β was a direct consequence of the higher antibody responses induced by the cVLP:IL-1β Q15G vaccine. However, the cVLP:IL-1β vaccine batch used for the dose-dependency study induced a similarly high IL-1β-specific Ab response as the cVLP:IL-1β Q15G vaccine, yet reduced IL-1β levels similar to the other cVLP:IL-1β vaccine batch (Appendix A vs. Figure 6c). This suggests that the higher IL-1β neutralizing capacity of the cVLP:IL-1β Q15G vaccine is somehow related to the introduction of the Q15G mutation, potentially affecting the specificity of the vaccine-induced Ab response.

IL-1β, as well as TNFα and IL-6 are pro-inflammatory cytokines, which are all important players in the host defense against microbial pathogens and viruses. Thus, effective neutralization of IL-1β could potentially lead to an increased risk of infection. Consequently, the effect of IL-1β depletion has been investigated in several studies. Of importance, prolonged use of Anakinra (IL-1R1 antagonist) in patients has not been associated with increased infection rates [51,52]. However, several studies in mice have shown an increased risk of infection in response to complete blockage of endogenous IL-1β, as well as in studies of IL-1R1-deficient mice [52,53,54,55,56,57,58]. Consequently, the effects of the cVLP:IL-1β and cVLP:IL-1β Q15G vaccines on IL-6 and TNFα levels were investigated. As expected, following DNFB challenge, both unvaccinated and vaccinated mice showed significantly increased IL-6 and TNFα levels compared to OOA-treated mice. This result verifies that other important pro-inflammatory cytokines remain present after the vaccine-induced neutralization of IL-1β, ensuring a backup in the case of bacterial infections. Furthermore, the significant reduction in clinical signs of ACD achieved by the IL-1β-targeted vaccination underlines that IL-1β, in contrast to IL-6 and TNFα, plays a central role in the inflammatory response to contact allergies. Finally, despite the limited sequence similarity between IL-1 family members, they share an overall three-dimensional fold [59,60,61]. Thus, to assess the specificity of the vaccine-induced Ab response, the reactivity of serum from mice vaccinated with cVLP:IL-1β or cVLP:IL-1β Q15G were tested against members of the IL-1 subfamily, consisting of IL-1β, IL-1α, IL-33, and the receptor antagonist, IL-1Ra. This analysis showed no cross-reactivity, suggesting that the vaccine-induced antibody response would not lead to off-target side effects.

Together, our results suggest the cVLP:IL-1β Q15G vaccine to be a promising vaccine candidate to be developed as an alternative treatment modality for severe ACD, providing a cheaper, safer, and more effective therapy for patients. These encouraging preclinical results support further investigation of the safety and efficacy of this vaccine in larger animal models. Moreover, as IL-1β serves as a universal pro-inflammatory cytokine, this vaccine could also have an effect on other IL-1β-driven diseases, as indicated by previous preclinical and clinical vaccination studies within RA and Type 2 Diabetes [27,28,29,30,62].

## Figures and Tables

**Figure 1 vaccines-10-00828-f001:**
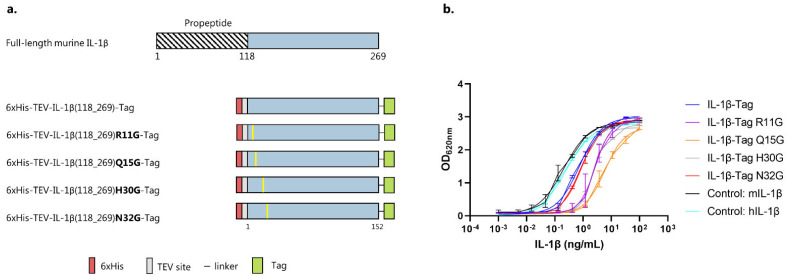
IL-1β antigen design. (**a**) Graphical representation of full-length (FL) murine IL-1β and the five IL-1β antigen constructs. FL murine IL-1β consists of a pro-peptide (aa1-117) that is cleaved to release the mature active 17kDa IL-1β protein (blue). The muIL-1β antigens were N-terminally fused to a 6xHis-tag (red), followed by a TEV site (gray). A split-protein binding Tag (green) was added to the C-terminus of the antigen, separated by a flexible linker. Mutations (yellow) were introduced into the WT sequence to reduce the affinity of the antigens for IL-1R1. (**b**) A HEK293 cell assay was used to assess the biological activity of the produced protein antigens. A dilution series of the IL-1β proteins (3-fold, starting from 100 ng/mL) was added to engineered HEK293 cells, and the biological activity was estimated based on SEAP detection (620 nm) after overnight incubation. Tested antigens include IL-1β-Tag (blue, EC_50_ = 0.75), IL-1β(R11G)-Tag (magenta, EC_50_ = 2.7), IL-1β(Q15G)-Tag (orange, EC_50_ = 5.8), IL-1β(H30G)-Tag (grey, EC_50_ = 2.4), IL-1β(N32G)-Tag (red, EC_50_ = 0.89). Positive controls include, muIL-1β (black, EC_50_ = 0.21) and huIL-1β (cyan, EC_50_ = 0.25). A technical replicate of this assay showed the same tendency (Appendix A). The EC50 values are shown in Appendix A.

**Figure 2 vaccines-10-00828-f002:**
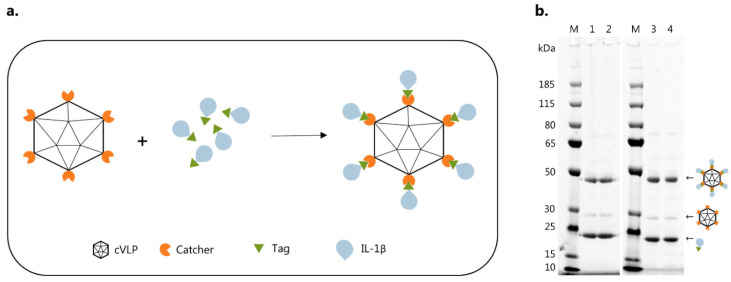
Vaccine formulation. (**a**) Schematic representation of the Tag/Catcher-AP205 technology used to create the cVLP:IL-1β vaccines. Genetic fusion of the split-protein Catcher to the AP205 capsid protein (total of 180 subunits per cVLP) allows for high-density, unidirectional display of IL-1β antigens fused to the corresponding binding Tag. (**b**) M: marker, lane 1: cVLP:IL-1β after O/N incubation at 4 °C (48 kDa), lane 2: cVLP:IL-1β after O/N incubation at 4 °C (48 kDa) + spin test, lane 3: cVLP:IL-1β Q15G after O/N incubation at 4 °C (48 kDa), lane 4: cVLP:IL-1β Q15G after O/N incubation at 4 °C (48 kDa) + spin test. Excess IL-1β-Tag antigen (21 kDa) and unconjugated cVLP subunits (27 kDa) were present in all the coupling reactions.

**Figure 3 vaccines-10-00828-f003:**
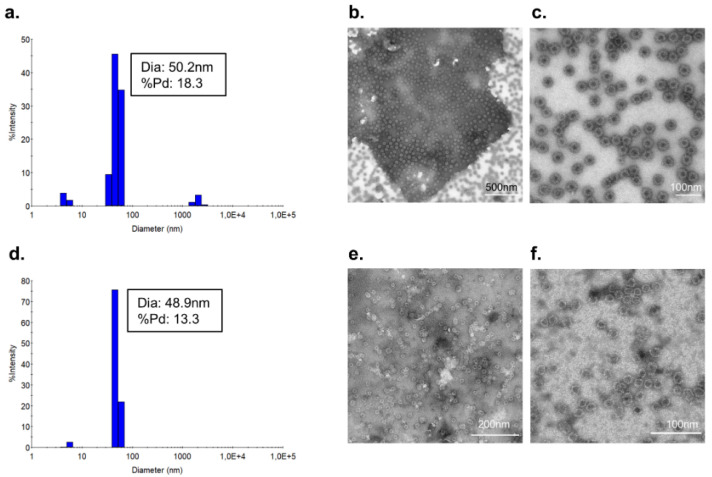
Vaccine quality assessment. Dynamic light scattering (DLS) analysis shows the % intensity of purified cVLP:IL-1β vaccines. The average diameter and polydispersity (% Pd) are annotated for each peak. Transmission electron microscopy (TEM) images of negatively stained purified cVLP:IL-1β vaccines. (**a**) DLS analysis of purified cVLP:IL-1β (**b**,**c**) TEM image of cVLP:IL-1β vaccine. (**d**) DLS analysis of purified cVLP:IL-1β Q15G. (**e**,**f**) TEM image of cVLP:IL-1β Q15G vaccine.

**Figure 4 vaccines-10-00828-f004:**
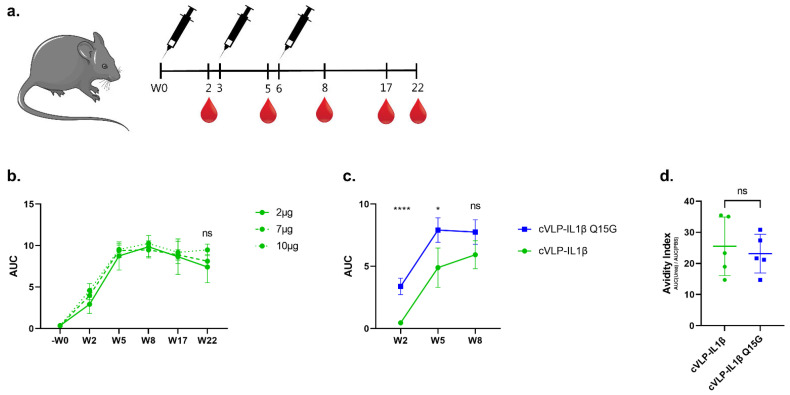
cVLP:IL-1β vaccines induce antigen-specific autoantibody responses in mice. (**a**) Schematic outline of the vaccine immunogenicity study, including time points for vaccinations and blood sampling. Serum samples were collected from female C57BL/6 mice two weeks after each immunization in a three-week interval prime-boost-boost (W0-W3-W6) regimen. For the initial antigen dose-escalation study, additional serum samples were collected at W17 and W22. ELISA results are depicted as the area under the curve (AUC) with mean ± standard deviation (SD). (**b**) Anti-IL-1β IgG titers measured before (−W0), and at different time points (weeks 2, 5, 8, 17, and 22) after vaccination are depicted as AUC values for groups of mice (*n* = 4) vaccinated with 2 µg (full line), 7 µg (dashed line) or 10 µg (dotted line) of cVLP-displayed IL-1β. (**c**) Anti-IL-1β IgG titers (AUC) measured in serum from mice (*n* = 5) vaccinated with the cVLP:IL-1β (green) and cVLP:IL-1β Q15G (blue) vaccines. The cVLP:IL-1β Q15G (blue) vaccine induces significantly higher IL-1β-specific IgG titers than the cVLP:IL-1β (green) vaccine at the week 2 (*p*-value > 0.0001) and week 5 (*p*-value = 0.0249) time points. Statistical analysis was performed on log-transformed values using one-way ANOVA, Tukey’s multiple comparisons test (adjusted *p*-value < 0.05 was accepted as significant, * < 0.05, **** < 0.0001). (**d**) An ELISA-based antibody avidity assay was performed on the serum from vaccinated mice. Avidity index values (%) are measured for serum samples obtained at week 8 after vaccination with cVLP:IL-1β (mean = 26 ± 9) and cVLP:IL-1β Q15G (mean = 23 ± 6). Statistical analysis was done using a non-parametric, two-tailed, Mann–Whitney test (adjusted *p*-value < 0.05 was accepted as significant).

**Figure 5 vaccines-10-00828-f005:**
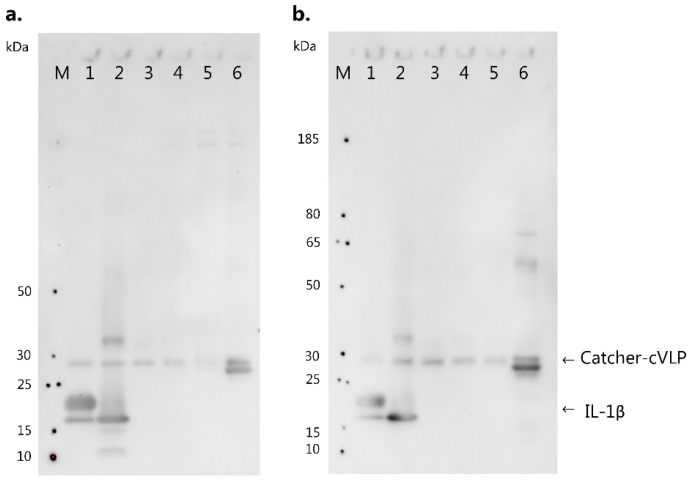
Vaccine-induced antibodies show no cross-reactivity with other IL-1 family members. Western blot analysis of the reactivity of cVLP:IL-1β vaccine-induced antibodies against members of the IL-1 subfamily, including IL-1β, IL-1α, IL-33 and IL-1Ra. M: marker, lane 1: 0.1 µg huIL-1β (17 kDa), lane 2: 0.1 µg muIL-1β (17 kDa), lane 3: 0.1 µg muIL-1α (17.5 kDa), lane 4: 0.1 µg muIL-33 (17.5 kDa), lane 5: 0.1 µg muIL-1Ra (17 kDa), lane 6: positive control of 0.1 µg Catcher-AP205 (27 kDa). (**a**) Reactivity of pooled serum from mice (*n* = 5) vaccinated with cVLP:IL-1β (**b**) Reactivity of pooled serum from mice (*n* = 5) vaccinated with cVLP:IL-1β Q15G. For all blots, goat-anti mouse IgG HRP-conjugated Abs was used as the secondary antibody.

**Figure 6 vaccines-10-00828-f006:**
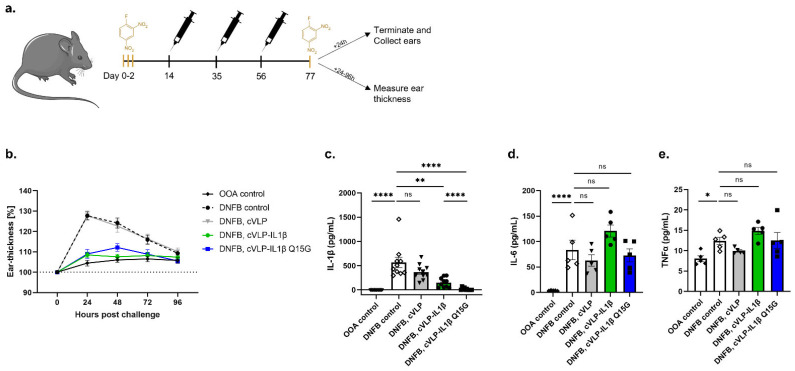
cVLP:IL-1β vaccination reduces ear thickness and IL-1β levels after DNFB challenge. (**a**) Schematic outline of the murine ACD challenge model. Mice were sensitized to the ears with 0.15% DNFB for three consecutive days, followed by a three-week prime-boost-boost immunization regime. The mice were subsequently challenged with 0.15% DNFB at the ears on day 77. The inflammatory response was followed in groups of mice, either by measuring the IL-1β level in inflamed ear tissue (top) or by measuring the ear thickness for up to 96 h post challenge (bottom). Graphs show data from two technical replicates (*n* = 5). (**b**) Ear thickness measured on groups of mice (*n* = 10) 24 h, 48 h, 72 h, and 96 h post-challenge, depicted as percentage (%) with mean ± SEM. (**c**–**e**) cytokine levels (i.e., (**c**) IL-1β, (**d**) IL-6 and (**e**) TNFα) with mean ± SEM measured in ear tissue from groups of mice (*n* = 5, two technical replicates) sensitized and challenged with (1) OOA, (2) 0.15% DNFB, (3) 0.15% DNFB and vaccinated with unconjugated cVLP, (4) 0.15% DNFB and vaccinated with cVLP:IL-1β and (5) 0.15% DNFB and vaccinated with cVLP:IL-1β Q15G. Statistical analysis was performed on log-transformed values using one-way ANOVA, Tukey’s multiple comparisons test (adjusted *p*-value < 0.05 was accepted as significant, * < 0.05, ** < 0.01, **** < 0.0001).

## Data Availability

The data that support the findings of this study are available upon request. Accession codes are the following, IL-1β protein (Gene ID: 16176), Acinetobacter phage AP205 coat protein (Gene ID: 956335).

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
