# Peer review of "Preclinical Efficacy of a Capsid Virus-like Particle-Based Vaccine Targeting IL-1β for Treatment of Allergic Contact Dermatitis"

_vaccines, 2022, doi:10.3390/vaccines10050828_

Round 1

Reviewer 1 Report

Status quo:

  • Interleukin-1β(IL-1β) is an important pro-inflammatory cytokine that has an effect onalmost every cell lineage in the body.
  • Clinical studies on the effect of IL-1β in ACD are lacking.

This preclinical study is of certain value.

Question:

1: L261 Figure 2. Vaccine formulation. The vaccine assembly experiment by SDS-PAGE is single, and further experiments are needed to verify the assembly, such as SPR or QCM-D and other methods, which can easily obtain their affinity assemblies

2:Figuer 6 The decreased level of IL-1β secretion indicated that neutralizing antibodies against IL-1β were elicited, and serum should be used to neutralize IL-1β in vitro instead of only measuring IL-1β levels. 

3:L155 Is the method of establishing the animal model original? If not you should cite references. Because it is not clear whether the model is established in the correct way. Or it should be shown in the introduction

Reviewer 2 Report

Goksøyr et al. Vaccines

Overall, it’s evident the authors have completed a thorough and informative analysis covering a prevalent area of concern affecting many individuals. There are some areas with grammar errors or omitted information, however, it is mostly acceptable. I found it perplexing, nevertheless, with the plethora of information obtained and observed, the authors could merely surmise two sentences for a conclusion.  There is ample information from the introduction, thorough methods, and plausible explanations in the results, hence, I had anticipated some form of solid opinions or ideas in the conclusion. What far reaching potentials would the authors think are conceivable and what is necessary to succeed? What is more, how likely would positive outcomes result, in their estimation? Unfortunately, we are left with a brief statement. It is unfortunate since I imagine the authors could discern more from the data than they have presented. However, specifics that do are listed below.

Page 1, line 43. List some side effect examples please.

Page 2, line 51. “No ACD treatment exist” should be either “treatments exist” or “treatment exists”.

Page 2, line 90. Please provide a reference for dialysis.

Page 3, line 100. Please provide information and/or reference for PCR thermocycler and thermocycler condition protocol.

Page 3, line 110. Please spell out “LPS”.

Page 3, line 113. What type of density gradient and reference for protocol?

Page 3, line 113. Centrifuge type, brand, other information is needed to repeat method.

Page 3, lines 123-124. How were the samples spun, and in what device?

Page 3, line 128. Were the TEM grids purchased with UA or was this protocol completed in the lab?

Page 3, lines 138, 142, 143, and others. Please note, sentences cannot be started using a numerical value “20, 0.5, etc.” The numbers must be spelled out “Twenty” or a different article used prior to the number. This happens several more times in the text. Please review.

Page 4, line 153. What volume of blood samples were acquired and where were they stored subsequently?

Page 5, lines 199-201. What software was used for the statistical calculations?

Figure 1. Please rotate to horizontal view, as is it is difficult to read, and the text is too small.

Figure 3. Please make the scale bars and measurements more legible, perhaps with a shadow text?

Figure 4 and 6. Same comment as Figure 1.

Page 11, lines 458, 460. Please refrain from referencing Figures within the text in the Discussion portion as these have previously been examined in the Results and this becomes redundant.

Page 12, lines 486-490. These two sentences just seem a bit inadequate for a conclusion. Please expand and give more in depth reflections to your analysis.

Figures S1-5. Please use various colors in all the charts for each aspect as it’s difficult to distinguish differences.
